# Improvement in Lactose Tolerance in Hypolactasic Subjects Consuming Ice Creams with High or Low Concentrations of *Bifidobacterium bifidum* 900791

**DOI:** 10.3390/foods10102468

**Published:** 2021-10-15

**Authors:** Gabriela Aguilera, Constanza Cárcamo, Sandra Soto-Alarcón, Martin Gotteland

**Affiliations:** 1Department of Nutrition, Faculty of Medicine, University of Chile, 8380453 Santiago, Chile; gabrielaguilera.o@gmail.com (G.A.); constanza.carcamoz@gmail.com (C.C.); 2Institute of Nutrition and Food Technology (INTA), University of Chile, 7830489 Santiago, Chile; sandra.soto.alarcon@gmail.com

**Keywords:** *Bifidobacterium bifidum*, probiotics, ice cream, hypolactasia, lactase, lactose tolerance, digestive symptoms

## Abstract

Although *Bifidobacterium bifidum* expresses lactase activity, no clinical trials have determined its impact on lactose-intolerant subjects. This study evaluated whether acute and chronic ingestion of ice creams containing *B. bifidum* 900791 at high (10^7^ CFU/g) or low (10^5^ CFU/g) concentrations improved lactose tolerance in hypolactasic subjects. Fifty subjects were selected based on a positive lactose (20 g) hydrogen breath test (HBT0) and the presence of digestive symptoms. The recruited subjects were required to perform breath tests after the acute ingestion of: (1) ice cream containing 20 g of lactose without a probiotic (HBT1); (2) the same ice cream, accompanied by a lactase tablet (HBT2); (3) the same ice cream containing the low or high dose of probiotic (HBT3-LD and HBT3-HD); and (4) after the chronic consumption of the ice cream without (placebo) or with the low concentration of probiotic for 1 month (HBT4). Significant decreases in H_2_ excretion during HBT2 and HBT3-HD as well as digestive symptoms during HBT2, HBT3-HD and HBT3-LD were observed compared to HBT0 and HBT1, while the orocecal transit time increased. Chronic consumption of the probiotic ice cream did not enhance lactose tolerance compared to the placebo. These results suggest that the acute ingestion of ice cream containing high or low concentrations of *B. bifidum* 900791 improves lactose tolerance in hypolactasic subjects.

## 1. Introduction

Milk is an important source of highly bioavailable calcium as well as protein of good nutritional quality. The main carbohydrate in milk, the disaccharide lactose, is hydrolyzed in the small intestine by the brush border enzyme lactase into glucose and galactose, which are subsequently absorbed [1]. Lactase activity is high in the newborn intestine but is genetically programmed to decrease from weaning, resulting in residual levels in adults. This situation, known as “primary hypolactasia” or “adult-type hypolactasia”, concerns around 75% of the world population and occurs earlier and faster in certain populations (such as the Thai population) than in others (such as the Finnish population) [1]. In Chile, it is estimated that 60–65% of the population is hypolactasic [2,3]. Primary hypolactasia is due to epigenetic events that result in the methylation of cytosine residues in the promoter area of the lactase gene. These DNA modifications inhibit the binding of transcriptional regulatory factors, leading to the progressive inhibition of lactase expression [4]. The remaining 25% of the world adult population is “lactase persistent”, i.e., adult intestinal lactase persists in these individuals in values similar to those of newborns. This phenomenon is due to specific mutations (single-base polymorphisms) which appeared in the human genome during the Neolithic age, the period in which human sedentarization, the domestication of cows, goats and sheep, and the beginning of dairy farming occurred [5]. Different mutations, which appeared at the same time in geographically distant populations (Northern Europe, sub-Saharan Africa, the Middle East, India and China) have been described, The presence of these mutations prevents the modification of cytosines, allowing the persistence of lactase gene expression throughout an individual’s entire life.

In hypolactasic subjects that consume milk, undigested lactose accumulates in the intestine where it can induce water secretion and eventually diarrhea, mainly due to its osmotic load [1]. Undigested lactose reaches the colon where it is fermented by the microbiota, generating gases such as hydrogen, which can cause bloating and abdominal pain, borborygmi and a higher frequency of rectal gas emission. In hypolactasic subjects, lactose can therefore be considered as a soluble dietary fiber. The eventual presence of digestive symptoms leads hypolactasic subjects to spontaneously reduce their consumption of milk and dairy products and, therefore, their intake of calcium and proteins of high biological value [1]. Although around 30% of hypolactasic individuals are lactose intolerant and develop digestive symptoms when they consume lactose-containing foods, most of them are tolerant and remain asymptomatic when consuming moderate amounts of milk (500 mL/d), probably due to the metabolic adaptation of their gut microbiota [6].

One way to continue consuming dairy products for intolerant people is, for example, to consume yogurt with live bacteria. Indeed, multiple clinical studies have shown that the lactase of yogurt bacteria continues hydrolyzing lactose in the human intestine, reducing the development of digestive symptoms [7]. Based on these studies, the European Food Safety Authority (EFSA) authorized a health claim stating that “the consumption of yogurt with live bacteria improves lactose tolerance in hypolactasic subjects” [8]. Similarly, many probiotics such as *L. acidophilus* NCFM express β-galactosidases that can hydrolyze lactose, preventing its subsequent fermentation and gas production [9,10]. Interestingly, a recent study in lactose-intolerant individuals reported that dietary supplementation with *L. casei* Shirota and *B. breve* Yakult for 4 weeks reduced their digestive symptoms and hydrogen excretion in a lactose hydrogen breath test performed upon completion of probiotic administration, an effect that persisted 3 months after discontinuing probiotic consumption [11]. Although *Bifidobacterium* spp. also express β-galactosidases [12,13] and could improve lactose intolerance, less clinical studies using this bacterial genus have been carried out in hypolactasic subjects, and none of them with *B. bifidum*. A limitation in the use of *Bifidobacterium* species is that they are oxygen-sensible, making it more difficult to reach and maintain high levels of these bacteria in food products [14]. 

Based on these antecedents, the aim of this study was to evaluate whether the acute and chronic intake of an ice cream containing high (10^7^ CFU/g) or low levels (10^5^ CFU/g) of the *B. bifidum* 900791 strain decreases breath hydrogen excretion and improves lactose intolerance in hypolactasic subjects. 

## 2. Subjects and Methods

### 2.1. Ethics and Subject Recruitment

The study protocol was approved by the Ethics Committee of INTA (Acta No. 2019-2, dated 20 March 2019), University of Chile, in compliance with the Helsinki Declaration. Each subject received detailed information about the aims and methods of the study, and those who agreed to participate and met the inclusion as well as exclusion criteria had to sign a written informed consent. The study protocol was registered in the international clinical trial registry database www.ClinicalTrials.gov (accessed on 30 August 2021), prior to subject recruitment (NCT03952988). 

Subjects auto-reporting digestive symptoms when consuming milk products were invited to participate through announcements published in the Faculty of Medicine and INTA. Recruited subjects were apparently healthy, 20 to 50 years of age, women and men. In addition, they had to carry out a basal hydrogen breath test (HBT0) with 20 g lactose dissolved in water in order to confirm their hypolactasic status and evaluate the presence of digestive symptoms. Exclusion criteria included pregnancy, chronic immune, tumoral and metabolic diseases, antecedents of digestive surgery or diseases including inflammatory bowel diseases and celiac disease. In addition, subjects with current or recent use of antibiotics, anti-inflammatories, laxatives or other drugs that interfere with intestinal transit and the microbiota were also excluded from the study. 

### 2.2. Basal Lactose Hydrogen Breath Test (HBT0)

Overnight-fasted subjects had to arrive at the Dpt. of Nutrition at 8:30 h. They drank 200 mL of mineral water with 20 g of lactose. As previously described [7], breath samples were obtained by end-expiratory sampling into 60 mL plastic syringes through a modified Haldane–Priestley tube. Two basal breath samples were obtained at a 10 min. interval before the lactose solution was consumed, and thereafter every 30 min for 3 h. The H_2_ content in the breath samples was determined with a breath analyzer (Lactotest, Medical Electronic Construction (MEC), Brussels, Belgium). Subjects with an increase in breath H_2_ higher than 20 ppm, compared to baseline, in three consecutive breath samples were considered as hypolactasic. Orocecal transit time (OCTT) was defined as the time at which a 20 ppm increase occurred. During the breath test, the presence of digestive symptoms (nausea, vomiting, abdominal pain, abdominal distension, increased rectal gas, borborygmi and diarrhea) was registered by the subjects, according to a semi-quantitative scale (0: absent; 1: mild; 2: moderate; 3: intense). 

### 2.3. Experimental Design—Study One

The study design is described in Figure 1. The first study evaluated the effect of the acute ingestion of probiotic-containing ice cream on lactose tolerance. With this aim, three HBTs were carried out with a one-week interval and in a randomized order: in HBT1, lactose in water was replaced by one serving of ice cream containing 20 g lactose, without *B. bifidum* 900791 (negative control). HBT2 was carried out after consuming one serving of ice cream with 20 g of lactose and without *B. bifidum* 900791, just after the ingestion of one tablet of exogenous lactase (Diolasa, Lab. Andromaco, Chile, 9000 FCC/tablet) as a positive control. HBT3-LD (low dose) and HBT3-HD (high dose) were carried out with one serving of ice cream containing 20 g of lactose with 10^5^ CFU/g or 10^7^ CFU/g of *B. bifidum* 900791, respectively.

### 2.4. Experimental Design—Study Two

The second study evaluated the effect of the chronic intake of probiotic-containing ice cream with a low concentration of probiotic (10^5^ CFU/g) on lactose tolerance. The 45 subjects were then randomly distributed into 2 groups who had to consume a daily serving of 50 grams of ice cream with (probiotic group) or without (placebo group) *B. bifidum* 900791 for four weeks. At the end of this period, every volunteer had to carry out a fourth HBT (HBT4) with the probiotic-free ice cream with 20 grams of lactose. During this period, every subject had to respond to a validated online survey, the Gastrointestinal Symptom Rating Scale (GSRS), which included 15 questions concerning digestive symptoms as well as stool frequency and consistency, which was weekly sent to them via Google Forms. 

### 2.5. Ice Cream Composition and Probiotic Determination

The ice cream used in this study was milk based and vanilla flavored. The ingredients used for its preparation were water, saccharose, glucose, stabilizer/emulsifier, skim milk powder, milk cream, liquid whole milk and vanilla pods. Its nutritional composition per serving was as follows: energy: 53 kcal; protein: 1.3 g; fat: 1.8 g; and carbohydrates: 8.0 g (including 1.4 g glucose, 3.1 g saccharose and 1 g lactose). For the determination of *B. bifidum* 900791, 20 g of ice cream was thawed, homogenized in sterile conditions with a stomacher and serially diluted in peptone water, with cysteine hydrochloride as a reducing agent [15]. One hundred microliters of the dilutions were spread on MRS agar plates that were subsequently cultivated in anaerobic conditions for 72 h at 37 °C. Counting was carried out in the plates containing between 30 and 300 colony forming units. The ice creams given to the subjects were freshly prepared to ensure that they contained the required density of bifidobacteria. In addition, we also confirmed that the levels of bifidobacteria remained stable over time in the ice cream. 

### 2.6. Statistical Analysis

The sample size was calculated using the decrease in the area under the curve (AUC) of hydrogen excretion obtained in HBT3 as the primary outcome. A decrease of at least 25% in AUC was expected in the treated group (with *B. bifidum* 900791) compared to the control. Considering an α value of 0.05 and a power of 80%, 22 subjects were required for each group. Considering a dropout rate of around 10%, 50 volunteers were finally recruited. 

Statistical analysis was carried out with the GraphPad Prism (version 5.0) software package (GraphPad Software, San Diego, CA, USA). Results were expressed as means ± SD or SEM. Digestive symptoms and OCTT in the different conditions were compared by a one-way ANOVA on ranks Kruskal–Wallis test and a post hoc Mann–Whitney U test with the Bonferroni correction. The AUCs of H_2_ excretions were compared by one-way ANOVA and a post hoc test.

## 3. Results

### 3.1. Flowchart of the Study

From the 77 subjects initially interested in participating in the study, 50 (64.9%) exhibited an increase in their breath H_2_ higher than 20 ppm compared to the basal values, and were classified as hypolactasic. All of them presented digestive symptoms of variable intensity during the test. These 50 subjects were recruited to participate in the study after they signed the informed consent. Seventy percent were female, the mean age of the participants was 28 ± 7 years (range: 21–50 y) and their body mass index 24.7 ± 3.1 kg/m^2^ (range: 18–36 kg/m^2^). The flowchart of the study is shown in Figure 2. Five volunteers abandoned the study (dropout ratio: 10%) during the first period (Study one): one after HBT0, two after HBT1, one after HBT2 and one after HBT3. Forty-five subjects were randomized into two groups to receive the probiotic-containing ice cream or the placebo ice cream for one month. Due to the SARS-CoV-2 pandemic, only 29 volunteers carried out HBT4 at the end of this period.

### 3.2. Acute Effect of Probiotic Ice Cream Intake on H_2_ Excretion

Changes in exhaled breath H_2_ during the different HBTs performed during the study and the corresponding areas under the H_2_ curves are shown in Figure 3A,B, respectively. The results of the initial HBT (HBT0) indicate that the intake of 20 g of lactose in water produced a strong H_2_ excretion. H_2_ values increased from minute 30, reaching a plateau at 50 ppm by 150 min, and were accompanied by an increase in digestive symptoms (Figure 4A,B). The ingestion of the same amount of lactose in ice cream (HBT1) caused a lower H_2_ excretion that reached a plateau at 38 ppm by 150 min, and a lower intensity of digestive symptoms compared to HBT0 (Figure 3A,B). In HBT2, volunteers had to consume the same ice cream with lactose, together with a commercial lactase tablet as a negative control. As expected, a lower H_2_ excretion was observed, which remained below 20 ppm during the entire test, with significant differences when compared to those of HBT0 and HBT1 from 90 and 120 min., respectively (Figure 3A,B). The intake of ice cream with low or high probiotic concentrations resulted in significantly less H_2_ excretion from 90 min onwards compared to HBT0. The excreted H_2_ values observed during HBT3-LD reached a plateau at 25 ppm by 120 min, but without significant differences compared to HBT1. However, contrarily to HBT1, H_2_ breath values for HBT3-LD did not differ from those of HBT2, except at 180 min. During HBT3-HD, the H_2_ excretion plateau was lower (19 ppm) and the H_2_ values significantly smaller than those of HBT1 at 150 and 180 min, and did not differ from the HBT2 values. Accordingly, the AUCs of H_2_ corresponding to HBT3-LD and HBT3-HD were significantly reduced compared to those from HBT0 and HBT1, but these decreases were only partial since the corresponding AUCs remained significantly higher than those described during HBT2 (Figure 3A,B).

### 3.3. Effect of Acute Ice Cream Intake on Digestive Symptomatology

During each HBT, volunteers were asked to record on a form the intensity (on a scale of 0 to 3) of seven gastrointestinal symptoms, as perceived. The sum of the intensity of each symptom could therefore vary between 0 and 21. Figure 4A shows the intensity of each of the different digestive symptoms perceived by the subjects in the different tests, and Figure 4B shows the sum of the intensity observed for each of the digestive symptoms. The symptoms recorded with the highest intensity during the HBTs were abdominal pain, bloating, borborygmi and increased rectal gas, while vomiting, nausea and diarrhea were the least reported or least intense symptoms, vomiting and diarrhea showing no significant differences between the different breath tests. 

In HBT0, all digestive symptoms tended to be more intense than in the rest of the tests, with bloating being the most intensely perceived symptom. In HBT1, a significant decrease in abdominal distension was observed compared to HBT0. In the presence of lactase (HBT2), the four main symptoms decreased significantly compared to those reported during HBT0 and HBT1. The intensity of abdominal distension and borborygmi as well as rectal gas decreased in HBT3-LD and/or HBT3-HD compared to HBT1 and HBT0, reaching the levels observed with exogenous lactase (HBT2). No differences were observed between HBT3-LD and HBT3-HD. Overall (Figure 4B), digestive symptomatology decreased with probiotic ice cream at both concentrations, reaching levels similar to those observed with the negative control (HBT2).

### 3.4. Effect of Acute Ice Cream Intake on Orocecal Transit Time

OCTTs determined for each HBT, except HBT2 (due to the lack of H_2_ increase in this condition) are shown in Figure 5A. The results indicate that OCTT did not vary between HBT0 and HBT1 (*p* = 0.065), while it increased in HBT-LD and HBT3-HD compared to HBT0 and tended to increase in HBT3-HD compared to HBT1 (*p* = 0.066).

### 3.5. Effect of Chronic Ice Cream Intake on H_2_ Excretion and Digestive Symptomatology

At the end of the first study, volunteers were randomized in two groups to receive a daily serving of the placebo or probiotic ice cream (low dose) for one month. HBT4 was carried out at this time in all the subjects. When the AUCs of breath H_2_ from both HBT4s were compared to HBT0 and HBT1 (Figure 3C), no significant differences were observed. Regarding digestive symptoms and stool frequency/consistency that were registered weekly by the subjects (GSRS survey) during the period of consumption, the only observed change was a higher rate of acid regurgitation in the probiotic group compared to the placebo group (*p* < 0.01), which was detected during the fourth week of product administration. No changes along time were observed for the different digestive parameters (data not shown). OCTT was also not affected by the intake of probiotic ice cream compared to the placebo (Figure 5B). 

## 4. Discussion

The aim of this study was to evaluate the effect of acute or chronic consumption of an ice cream containing *B. bifidum* 900791 on lactose tolerance in hypolactasic subjects. Ice creams are currently considered a good support for probiotic strains, and *Lactobacillus* spp. and *Bifidobacterium* spp., including *B. bifidum,* have shown good survival and maintain their ability to resist gastrointestinal conditions when added to this dietary matrix [16,17]. Food regulations in Chile and other countries generally state that probiotic foods should contain at least 10^7^ CFU/g of product at the time of consumption. However, reaching this concentration and maintaining it during the shelf life of the food is more difficult with oxygen-sensitive bacteria such as *Bifidobacterium* spp. Accordingly, bifidobacteria counts in commercial products are frequently low or even undetectable [14]. It is therefore interesting to evaluate the possibility of observing healthy effects with lower concentrations of bifidobacteria. For this reason, ice creams with high and low concentrations of *B. bifidum* 9000791 were compared in the present study. Interestingly, though less H_2_ excretion was only reported with the ice cream containing the higher probiotic content, our results indicate that digestive symptoms were significantly decreased not only with this product, but also with the ice cream containing the low *B. bifidum* concentration.

Subjects were recruited based on the results of an HBT (HBT0) performed with 20 g of lactose in water. About 65% of the participants showed H_2_ increases that, on average, reached a plateau at 50 ppm, accompanied by moderate digestive symptoms. This proportion of hypolactasic individuals in this group of subjects was similar to that reported in previous studies in Chile [2,3]. When the same test was performed with ice cream containing the same amount of lactose (20 g), a non-significant decrease in H_2_ excretion was observed, accompanied by a significant reduction in abdominal distension. These results suggest that less lactose reaches the colon and less fermentation and gas production occur. This is probably due to the fact that ice cream has a higher caloric density, a factor well-known to slow gastric emptying [18]. Consequently, this would reduce the amounts of lactose reaching the intestine per unit time, making it more likely to be hydrolyzed by the residual lactase and therefore improving lactose digestion in these subjects. Slowing gastric emptying is, in fact, an important strategy for improving lactose tolerance in lactose-intolerant subjects [19]. In HBT2, subjects consumed the ice cream with lactose together with a commercial exogenous lactase tablet. As expected, a strong reduction in their H_2_ excretion and digestive symptoms was reported, confirming therefore that exogenous lactase improves lactose tolerance in hypolactasic subjects [20]. When the volunteers performed the third breath test (HBT3) with the ice cream containing lactose and the probiotic strain (10^7^ CFU/g), a significant decrease in exhaled H_2_ levels was observed, while only a non-significant decrease was observed with the same product containing 10^5^ CFU/g. In a recent review on the beneficial effects of various strains on lactose intolerance, significant effects were reported with probiotic concentrations ranging between 10^8^ and 10^11^ CFU/g [21]. However, yogurt is known to improve lactose tolerance with bacterial concentrations (*L. bulgaricus* and *S. thermophilus*) of 10^7^ CFU/g [8]. It is likely that the dose at which a probiotic exerts its lactose-tolerance-enhancing effect depends on the strain and its ability to produce lactase activity. Therefore, the benefit of *B. bifidum* 900791 on H_2_ excretion is dose-dependent, and the lowest dose was insufficient to reduce this parameter in our study.

Regarding digestive symptoms, some authors have described that *L. acidophilus* in milk attenuated symptoms at concentrations of 10^9^ but not 10^8^ CFU/mL, while *L. bulgaricus* reduced exhaled H_2_ and digestive symptoms at both concentrations [22]. In our study, it is interesting to note that the ice cream with the low concentration of probiotic was able to reduce the intensity of the main digestive symptoms, those being abdominal pain, bloating and increased rectal gas. Based on these observations, one may wonder how low-dose probiotic ice cream can improve symptomatology with only mild effects on H_2_ excretion. Our results show that the presence of the probiotic in ice cream is associated with a slower OCTT compared to that observed in lactose in water or in ice cream. This phenomenon could reduce the osmotic load associated with the presence of lactose in the lumen, favor water reabsorption (reducing borborygmi and eventually abdominal pain as well as osmotic diarrhea) and decrease the inward flow of lactose into the colon, positively affecting gas production and abdominal distention. Therefore, the faster the OCTT, the more intense the digestive symptoms observed will be after an acute lactose load. For the same reason, people with lactose intolerance generally have a faster orocecal transit [23]. However, no correlations were found between digestive symptoms and OCTT, perhaps due to the large interindividual variability in symptom perception. It should be noted that most studies evaluating the effect of probiotic strains on OCTT in humans have reported a shortening of this parameter, most particularly in subjects with constipation and, to a lesser extent, in healthy subjects [24]. However, other strains were reported to reduce bowel movements, increasing intestinal transit time and decreasing loose stools in patients with diarrhea-predominant irritable bowel syndrome [25,26]. It is therefore probable that *B. bifidum* 900791 contributes to restore the OCTT when it is accelerated by lactose intake in hypolactasic subjects.

The second part of the study was conducted to evaluate whether the chronic consumption of probiotic ice cream could improve lactose tolerance in lactose-intolerant subjects compared to the consumption of the same ice cream without the probiotic (placebo). This objective was supported by the study of Casuccio et al. [11], who reported that a 4-week supplementation with *L. casei* Shirota and *B. breve* Yakult in lactose-intolerant subjects reduced their digestive symptoms and hydrogen excretion in a lactose HBT performed after the completion of probiotic administration, an effect that persisted 3 months after probiotic consumption was discontinued. The explanations for such effect are unclear. Studies evaluating the effect of chronic probiotic ingestion on lactose tolerance in hypolactasic subjects have given contradictory results. A 2-week consumption of *L. acidophilus* B62F04 [27] or *L. acidophilus* DDS-1 [28] improved abdominal symptoms compared to a placebo. Symptom improvement without a parallel decrease in H_2_ excretion was described after a 6-month supplementation with 11 different probiotic strains [29]. Similar positive results for bloating and constipation were observed with *L. reuteri* [30], *B. longum* and *B. animalis*, as well as for *B. longum* BB536, *L. rhamnosus* HN001 and vitamin B6 [31,32]. In contrast, a 6-week supplementation with *B. animalis* IM386 and *L. plantarum* MP2026 did not report any significant differences in H_2_ excretion and digestive symptoms compared to the control group [33].

Using a similar design, we could not confirm these results, as no differences in H_2_ excretion during HBT4 and in the digestive symptoms along the treatment period were reported between the placebo and probiotic groups. However, it is possible that our study was underpowered because of the high dropout rate of the volunteers due to the SARS-CoV-2 pandemic and its subsequent premature end, as described for several other studies [34]. Symptoms were generally mild and infrequent, with unclear variability between each week and group. Another limitation of our study was that the diet of the participants and more particularly their intake of fibers or lactose-containing foodstuffs was not controlled during the treatment period and may have had an impact on digestive symptoms. On the other hand, both ice creams contained small amounts of lactose (as milk powder) which could have favored a certain adaptation of the microbiota to this disaccharide, reducing the differences in H_2_ excretion and digestive symptoms between both groups [6]. It is possible that a significant effect could be observed with ice cream containing a higher concentration of probiotics.

## 5. Conclusions

Until now, there were no data available on the use of *B. bifidum* for the nutritional management of lactose-intolerant individuals. This study therefore provided new results supporting such use in this population. This study also confirms ice cream as an adequate dietary matrix for the incorporation of probiotics, and more particularly those from the *Bifidobacterium* genus. Finally, our study suggests that ice cream containing low concentrations (10^5^ CFU/g) of this probiotic strain are also successful in improving lactose tolerance in hypolactasic subjects.

## Figures and Tables

**Figure 1 foods-10-02468-f001:**
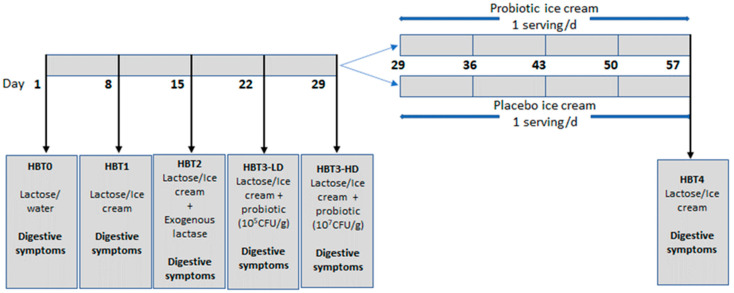
Study design.

**Figure 2 foods-10-02468-f002:**
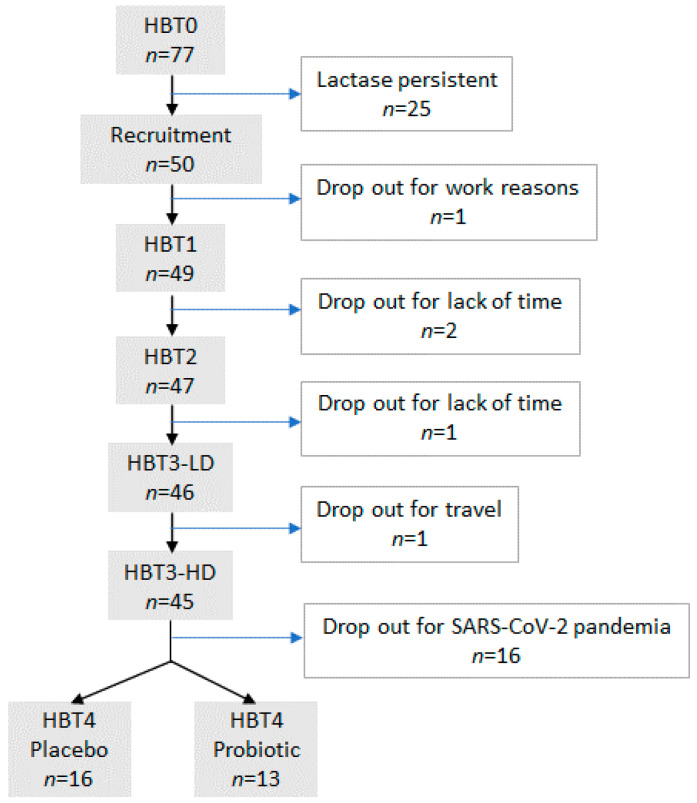
Flowchart of the study.

**Figure 3 foods-10-02468-f003:**
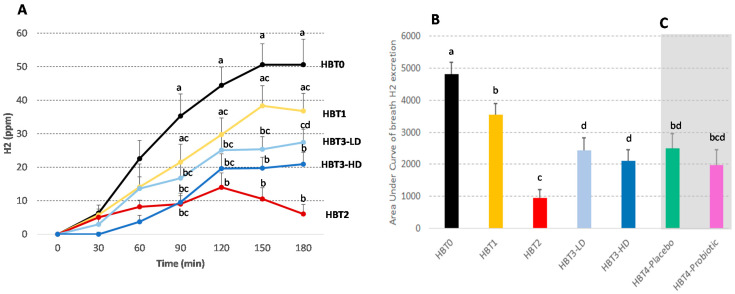
Changes in breath H_2_ excretion during the different HBTs carried out during the study. (**A**) Changes in respiratory H_2_ excretion (ppm) during HBT0 (black), HBT1 (yellow), HBT2 (red), HBT3-LD (light blue) and HBT3-HD (dark blue). (**B**) Area under the curve (AUC) of H_2_ excretion during the different breath tests carried out to determine the effect of the acute or (**C**, grey area) chronic intake of the probiotic ice cream. In (**A**), values with different letters (a, b, c, d) for the same time are significantly different (*p* < 0.05). In (**B**,**C**), bars with different letters (a, b, c, d) are significantly different (*p* < 0.05). Means +/− SEM.

**Figure 4 foods-10-02468-f004:**
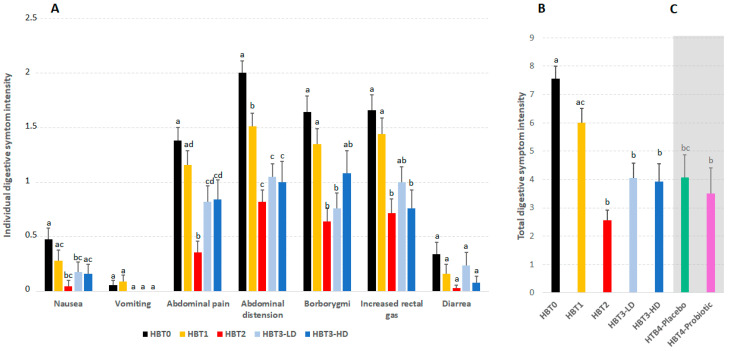
Digestive symptoms during the acute and chronic intake of the ice creams. (**A**) Intensity of the individual digestive symptoms during the acute study, corresponding to the different hydrogen breath tests (HBTs). (**B**) Intensity of the total digestive symptoms during the acute intake of the ice creams and (**C**, grey area) during the chronic intake in the placebo and probiotic groups. Means +/− SEM. Results between groups were compared by a one-way ANOVA on ranks Kruskal–Wallis test and a post hoc Mann–Whitney U test with the Bonferroni correction. For each digestive symptom shown in **A**, bars with different letters (a, b, c, d) are significantly different (*p* < 0.05). In (**B**,**C**), bars with different letters (a, b, c) are significantly different (*p* < 0.05).

**Figure 5 foods-10-02468-f005:**
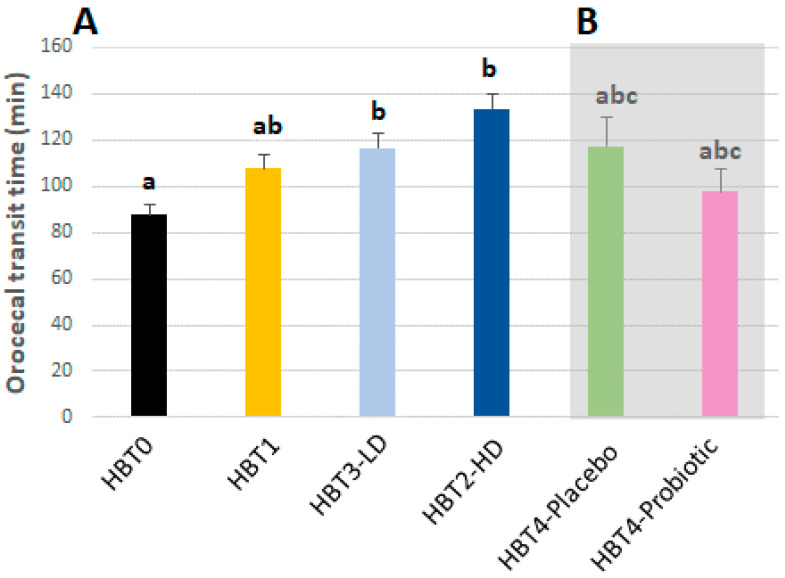
Orocecal transit time (OCTT). OCTTs were determined during each HBT from the studies with acute (**A**) and chronic (**B**) intake of the ice creams. Means +/− SEM. Results between groups were compared by a Kruskal–Wallis test and a post hoc Mann–Whitney U test with the Bonferroni correction. Bars with different letters (a, b, c) are significantly different (*p* < 0.05).

## Data Availability

Not applicable.

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
