# Peer review of "Improvement in Lactose Tolerance in Hypolactasic Subjects Consuming Ice Creams with High or Low Concentrations of Bifidobacterium bifidum 900791"

_foods, 2021, doi:10.3390/foods10102468_

Round 1

Reviewer 1 Report

The paper by Aguilera et al on “Improvement of Lactose Tolerance in Hypolactasic Subjects 2 Consuming Ice Creams with High or Low Concentrations of 3 Bifidobacterium bifidum 900791” is an interesting one. Only a few considerations have to be made about the current manuscript:

  1. Line 166. The word Recutamiento should be given in English
  2. Line 172: what is the reason for the predominant number of females compared to men?
  3. Line 203: Are the H2- in values in Figure 1 A given as SD or SEM?
  4. Methods: In the method section it is only declared that ice cream was used as carrier for the Bifidum bacteria. What kind of ice cream? Milk based? What taste? The concentration of carbohydrates was 9%. What kind of sugars: Lactose, sucrose, glucose or fructose?

Was there a group of subjects fed only ice-cream without additions and consecutive H2 breath test? This probably would have been the proper control group.

  1. If the chronic consumption of ice cream with probiotics does not influence the digestive symptoms in lactose intolerant persons, the question must be allowed, what practical consequences that finding has, because then ice cream with probiotics would not be a normal foodstuff to be recommended for daily of frequent use?

Author Response

We thank the reviewer for his comments and suggestions.

1- The word “reclutamiento” was corrected in Figure 2.

2- The fact that more women were recruited than men is only because more women were interested in participating in the study. Although the prevalence of hypolactasia is the same in men and women, women tend to complain more frequently of lactose intolerance, perhaps because they tend to suffer more from irritable bowel syndrome and think (probably wrongly) that this condition is related to lactose intolerance.

3- H2 values were given as SEM (as specified in the legend of the figure 3)

4- More data about the type of ice cream and the ingredients used in its preparation were given in the text of the Method section (L156-160).

We agree with the reviewer that it would have been interesting to perform a BHT after ingestion of a bifidobacteria-free, lactose-free ice cream. However, the addition of a BHT would have been too burdensome for the participants and we felt that ingestion of ice cream with lactose, simultaneously with an exogenous lactase, was a better control because it allowed us to compare the efficacy of the probiotic with that of the enzyme.

5- Although chronic consumption of ice cream with probiotics does not influence the digestive symptoms of lactose-intolerant subjects, it is interesting to see that acute consumption of this product manages to reduce this symptomatology, even when this probiotic is present in low concentrations. This supports the use of this strain in other dairy products or in lyophilized form, together with these products, to allow the intolerant subjects to consume them without having undesirable effects.

Reviewer 2 Report

  1. It would be good if the authors could repeat the experiment, this time ensuring all patients being under the same conditions (in terms of food, water, probiotics, etc.).
  2. It would be good to determine the number of viable bifidobacteria cells after their incorporation in the ice-cream mass as well as right before consumption.
  3. It would be good to include 2-3 bifidobacteria strains in the examination.

Author Response

1- In this study, the volunteers were asked to avoid the consumption of other products with probiotics and for this purpose, they were given a list of these products available in the Chilean market. Regarding dietary consumption, the fact that each subject is his or her own control (in the acute consumption study) helps to reduce the potential variability associated with this factor.

2- Before starting the study on the volunteers, we confirmed that the levels of bifidobacteria in the ice cream remained stable over time. In addition, the ice creams given to the subjects were freshly prepared to ensure that they contained the required density of bifidobacteria.

A sentence about this matter was added in the text of the Material section (L165-68)

3- Since the probiotic properties are strain-specific, the inclusion of several strains of Bifidobacterium in the product would not allow us to determine which strain or strains, specifically, improve lactose intolerance in these subjects. In addition, since no other study used B. longum in this type of subject, it was important to test one strain and not a mixture of strains. However, it would be interesting to evaluate the use of combined strains in future studies.

Reviewer 3 Report

General comments: The authors studied the effect of probiotic supplementation in ice cream on improvement of lactose tolerance. The study design was appropriate. The results were thoroughly analyzed and discussed. The manuscript in general was well written except for a few syntax errors.  Additional queries, comments, and suggestions can be found in my specific comments.

Specific comments

Line 24: Change “placebo” to “the placebo”.

Line 103: Confusing sentence.

Line 104: Change “ml” to “mL” throughout the manuscript.

Line 124: Why did the authors want to test the low dosage since probiotic food should contain at least 10^7 CFU/g of product?

Lines 156-157: Awkward sentence. Please revise.

Line 162: Change “one-way” to “one-way ANOVA”.

Line 163: Please specify when an ANOVA or Kruskal-Wallis test would be used?

Line 163: Was the Mann-Whitney U-test only used as the post hoc test for Mann-Whitney U-test, or was it also used for ANOVA?

Line 164: Please specify which statistical software was used for power analysis and difference tests.

Line 173: The unit for BMI is incorrect.

Line 246: Delete “co”.

Lines 259-262: Long and awkward sentence. Please rephrase.

Author Response

1- Line 24: Change “placebo” to “the placebo”.

"placebo" was changed to "the placebo".

2- Line 103: Confusing sentence.

The sentence was improved

3- Line 104: Change “ml” to “mL” throughout the manuscript.

ml was changed to mL elsewhere in the text.

4- Line 124: Why did the authors want to test the low dosage since probiotic food should contain at least 10^7 CFU/g of product?

Although probiotic foods should contain at least 10^7 CFU/g of product, achieving this concentration and maintaining it over the shelf life of the food is more difficult with oxygen-sensitive bacteria such as Bifidobacterium spp. As shown in ref. 14 (just added in the text), a high proportion of commercial products claiming to contain bifidobacteria actually have none or very low concentrations, clearly less than 107 CFU/g.   It is therefore interesting to see whether lower concentrations may also have health effects. 

5- Lines 156-157: Awkward sentence. Please revise.

The sentence was rewritten. (L170-71)

6- Line 162: Change “one-way” to “one-way ANOVA”.

The sentence was corrected according to the reviewer’s suggestion. (L173)

7- Line 163: Please specify when an ANOVA or Kruskal-Wallis test would be used?

The corresponding sentence was rewritten to specify the use of ANOVA or KW test (L176-180).

8- Line 163: Was the Mann-Whitney U-test only used as the post hoc test for Mann-Whitney U-test, or was it also used for ANOVA?

No, the Mann-Whitney U test was used as post-hoc test only with the Kruskal-Wallis test

9- Line 164: Please specify which statistical software was used for power analysis and difference tests.

This data was added in the text L176

10- Line 173: The unit for BMI is incorrect.

BMI unit was corrected (L188)

11- Line 246: Delete “co”.

The sentence was corrected (L272)

12- Lines 259-262: Long and awkward sentence. Please rephrase.

The sentence was simplified

Round 2

Reviewer 3 Report

The authors have addressed all my comments and revised the manuscript accordingly. I have only one minor comment: in line 181, please specify which post hoc test was used.